# DNA metabarcoding uncovers the diet of subterranean rodents in China

**Xuxin Zhang, Yao Zou, Xuan Zou⬚, Zhenggang Xu⬚, Xiaoning Nan¤\*, Chongxuan Han⬚¤\***

Key Laboratory of National Forestry and Grassland Administration on Management of Western Forest Bio-Disaster, College of Forestry, Northwest A & F University, Yangling, Shaanxi, China

¤ Current address: Yangling, Shaanxi, China
\* 358727493@qq.com (XN); 1139186993@qq.com (CH)

**Data Availability Statement:** The raw reads were deposited into the NCBI Sequence Read Archive (SRA) database 169 (Accession Number: PRJNA753876).

## Abstract

### Objective

A type of rodent called a zokor causes great harm to agriculture and forestry production due to its large and sophisticated diet. As this rodent lives subterrane for most of its life, researchers know little about its dietary habits. Further understanding of its diet is important for developing green and sustainable control strategies for the zokor.

### Methods

Chloroplast *trn*L gene and internal transcription spacer 1 primers were selected for high-throughput sequencing of stomach contents of captured zokor by DNA metabarcoding.

### Results

A total of 25 zokors were selected, the food list of 32 families, 80 genera, and 154 species was obtained. At the family level, it was found that zokors mainly fed on Asteraceae, Poaceae, Rosaceae, Pinaceae, Brassicaceae, and Apiaceae. At the genus level, zokors mainly fed on *Echinops*, *Littledalea*, *Artemisia*, *Picea*, *Cirsium*, and *Elymus*. The diet alpha diversity of *Eospalax cansus* was slightly higher than that of *Eospalax cansus* (*P*>0.05). The zokor's diet is highly phconsistent with the resources of its habitat. Most food choices tend to be the same between the two zokors. They fed primarily on *Calamagrostis*, *Cirsium*, *Echinops*, *Medicago*, *Sanguisorba*, and *Taraxacum*. Zokors mainly fed on the roots of perennial herbs, which are important source of energy.

### Conclusion

High-throughput sequencing-based DNA metabarcoding technology has effectively revealed the diet of zokors and indicated that zokors are food generalists.

**Funding:** This study was supported by Ecological Adaptation and Molecular Basis of spreading epidemic of Major biological hazards in Forest (2017YFD0600103–4-3), a National Key R & D project of Ministry of Science and Technology, PRC. The funders had no role in study design, data collection and analysis, decision to publish, or preparation of the manuscript.

**Competing interests:** The authors have declared that no competing interests exist.

## Introduction

Zokor(Myospalacinae) is a kind of herbivorous subterranean rodent endemic in East Asia [1–3]. They generally have well-developed forelimbs suitable for digging, good senses of smell and hearing, but poor vision. These rodents require a lot of energy to dig and maintain a complex system of tunnels, especially in hard, dry soil [4]. This has forced zokors to expand their food foraging [2, 4]. Animals that forage underground, with large search costs and diverse food resources, must collect all edible food species found when digging burrows. This, combined with the non-directed search pattern, will produce a generalist foraging behaviors [5].

There are currently nine species of zokor in China, including six species of *Eospalax* and three species of *Myospalax* [3]. They do harm to the industries of new woodland [6, 7], crops, and herbs [8, 9] in different degrees. The *E. smithii* and *E. cansus* inhabiting in the south of Liu-p'an Mountains in loess Plateau seriously harm the newly cultivated forest. The average daily food intake of *E. cansus* is 105.7 g of herbaceous plants and 60.5 g of tree roots [7]. Domestic research on zokors began out of a desire to prevent and control of zokor-related damage in China. It is important to explore the dietary characteristics of zokor for formulating scientific rodent control measures [10].

To date, traditional methods are used to study the diet of zokors, such as by observing captive feeding [11], food accumulation in caves [12], and foraging behavior and microhistology of gastric contents [13, 14]. The aforementioned methods are limited in terms of season and food diversity, and can only reflect the dietary traits of zokors in particular time periods. Microhistology is the advanced technique in the current study of zokor diet. This method requires accurate analysis of partially digested plant fragments but may overemphasize indigestible plants [15]. Furthermore, it is limited by the researchers' taxonomic ability of plants and imposes a heavy workload, and these factors cause the study on zokor diet to be incomprehensive and unobjective.

Over the past few years, with the improvement of molecular biology and the emergence of high-throughput sequencing (HTS), DNA metabarcoding was developed, which integrates DNA barcoding and HTS. DNA metabarcoding is not restricted by prey species and can identify prey at the taxonomic species level, so it is uniquely suitable for diet research in difficult-to-observe or special habitats. Moreover, it allows for parallel processing and sequencing of large samples [16]. DNA metabarcoding mainly uses universal primers to amplify food DNA fragments of animals; hence, the selection of primers is particularly crucial. The combination of three primers can improve the food variety by 30% [17]. The use of the *trn*L (UAA) P6 loop combined with ITS1-F/ITS1*Poa*-R and ITS1-F/ITS1*Ast*-R can improve the species resolution of Poaceae and Asteraceae [15]. Therefore, it is vital to combine different types of primers for the study of the diet of zokors, which is a broad-feeding animal.

With the deepening of the concept of ecological development, the control of zokors has shifted from pesticide poisoning and physical killing to the concept of protecting selected tree species [18]. This study focuses on solving the following problems: (1) Whether DNA metabarcoding can be used to study the diet of zokors, a kind of subterranean rodent? If so, how much better than traditional methods? (2) What is the food composition of the two zokor species distributed in Liu-p'an Mountains? (3) What is the significance of the application of DNA metabarcoding technique in the study of feeding habits of zokor for the prevention and control of zokor?

## Materials and methods

### Study area and flora

The study area was located in Longde County, Guyuan City, Ningxia Hui Autonomous Region, China, the average altitude is 2250 m. We selected seven regions with an area of 10,000

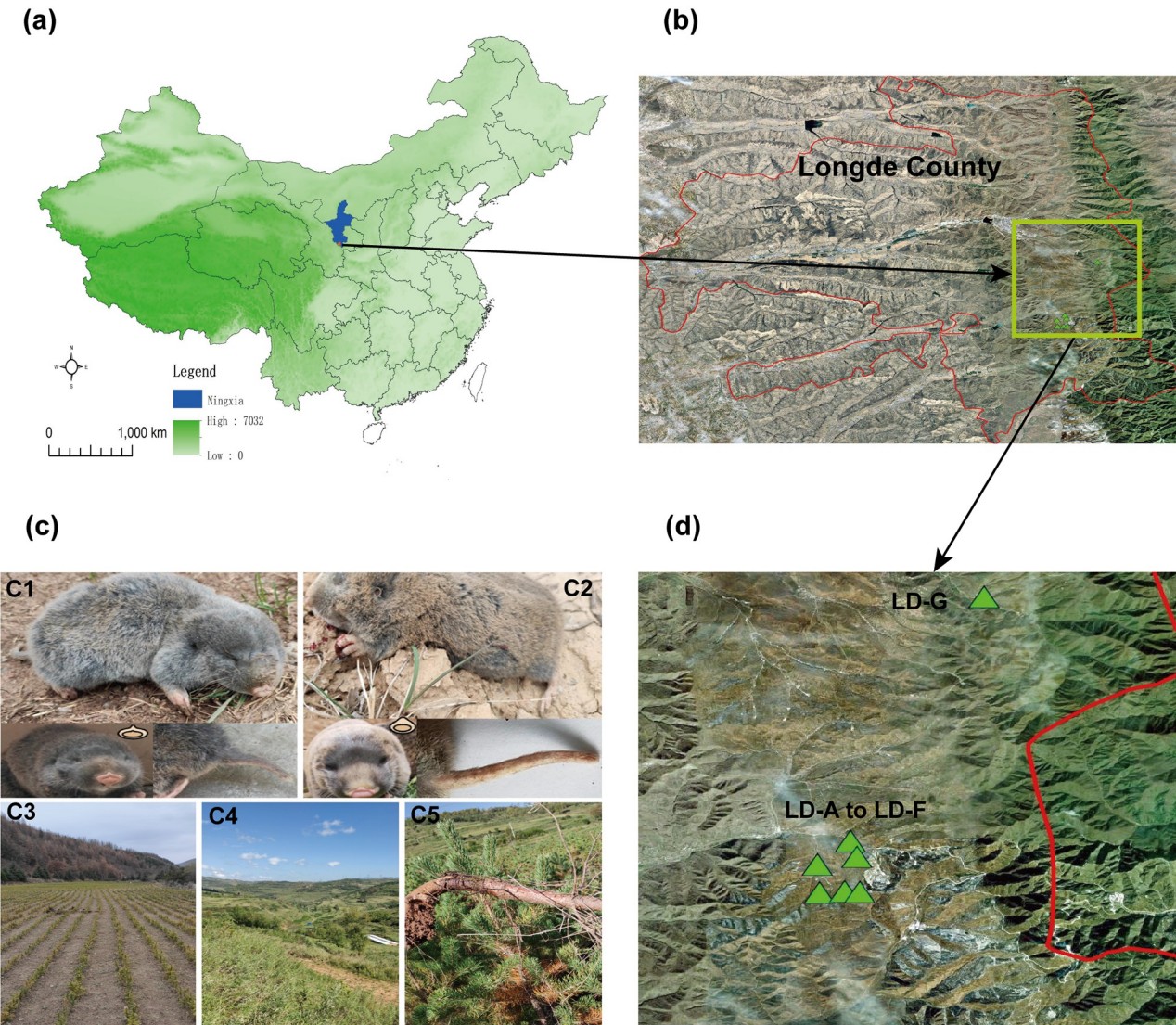

**Fig 1. Location and habitat diagram of the study area and map morphological contrast of two zokor species. a**. The location of the study area in Ningxia Hui Autonomous Region, China. **b**. The red line is the county boundary and the green rectangle is the approximate scope of the study area. **c**. Comparison of the morphology of two species of zokor. **1c1**. *E. smithii*, coat is taupe, has a short tail with bushy short hairs and an upward spike on the snout; **c2**. *E. cansus*, coat is tawny, has a long tail with sparse short hairs and a smooth upward projection on the snout; **c3**. LD-G habitat map; **c4**. LD-A to LD-F habitat map.**c5**. A root gnawed by the zokors. **d**. Satellite map of the seven study areas, with green triangles representing study sites.

square meters, and all of these regions were distributed by *E. cansus* and *E. smithii*. The two species of zokor have a habitat range of about 3 million square meters (Fig 1). The study area is a project area for of returning farmland to forest and grassland, in which the main tree species were *Pinus tabuliformis*, *Pinus sylvestris*, *Picea crassifolia*, and *Hippophae rhamnoides*. The main herbaceous plant genera were *Elymus*, *Leymus*, *Artemisia*, *Cirsium*, and *Bupleurum*.

## Acquisition of stomach contents from zokor

As zokors are a local pest, our samples came from a "rat hunt" conducted by the local forestry department. We obtained the wild zokors with the approval and assistance of the Forest Pest Control station of the Natural Resources Bureau of Longde County, Ningxia Hui Autonomous

**Table 1. Specimen information.**

| Plot ID | Altitude/m | Longitude and latitude | Specimen ID | Date/Year.month.day | Gender | Weight/g |
|---------|-----------|------------------------|-------------|---------------------|--------|----------|
| LD-A | 2260 | 106.1670 E; 35.5105 N | EC-1 | 2020.10.3 | female | 214.0 |
| | | | EC-7 | 2020.10.25 | female | 174.5 |
| | | | ES-4 | 2020.10.25 | female | 187.9 |
| | | | ES-13 | 2020.10.26 | female | 197.6 |
| | | | ES-14 | 2020.10.26 | female | 185.4 |
| LD-B | 2271 | 106.1683 E; 35.5065 N | ES-5 | 2020.10.25 | male | 242.7 |
| | | | ES-6 | 2020.10.25 | female | 208.2 |
| | | | ES-7 | 2020.10.25 | male | 357.2 |
| LD-C | 2268 | 106.1603 E; 35.5046 N | ES-1 | 2020.9.3 | male | 355.7 |
| | | | ES-8 | 2020.10.25 | male | 307.5 |
| | | | ES-9 | 2020.10.25 | female | 171.7 |
| LD-D | 2332 | 106.1607 E; 35.4976 N | ES-2 | 2020.9.3 | male | 226.4 |
| | | | ES-10 | 2020.10.25 | female | 221.4 |
| | | | ES-11 | 2020.10.25 | female | 308.8 |
| LD-E | 2242 | 106.1658 E; 35.4978 N | EC-5 | 2020.10.3 | male | 357.0 |
| | | | ES-3 | 2020.9.3 | male | 267.7 |
| | | | ES-12 | 2020.10.25 | female | 220.0 |
| LD-F | 2242 | 106.1688 E; 35.4977 N | EC-2 | 2020.10.3 | male | 270.0 |
| | | | EC-3 | 2020.10.3 | male | 358.0 |
| | | | EC-4 | 2020.10.3 | male | 352.0 |
| | | | EC-6 | 2020.10.3 | male | 327.0 |
| LD-G | 2262 | 106.1943 E; 35.5719 N | EC-8 | 2020.10.26 | female | 226.7 |
| | | | EC-9 | 2020.10.26 | male | 337.7 |
| | | | ES-15 | 2020.10.26 | male | 338.1 |
| | | | ES-16 | 2020.10.26 | female | 238.6 |

Note: EC and ES here and later stand for *E.cansus* and *E.smithii* respectively.

Region, China, and the study area was not a nature reserve. During the "rat hunt," rodents were trapped in ground arrow traps, which induced unconsciousness within a short time of being hit to minimize pain. The harvesting of zokors and the collection of stomach contents was approved by the Forest Pest Control Station in Longde County. All experiments were approved and supervised by the Science Ethics Committee of Northwest A & F University. Dissection was performed after recording relevant body indexes of zokors. The digestive tract was stripped out of each zokor, and the stomach contents were clamped with sterilized scissors and transferred to a 2 mL cryopreservation tube. After collection, the tubes were immediately stored in liquid nitrogen gas phase for subsequent experiments. Among the captured zokors, 25 were selected for diet analysis. Among them, 16 were *E.smithii* (8 male, 8 female) and nine were *E.cansus* (6 male, 3 female) (Table 1).

## DNA barcode selection

DNA extracted from stomach contents of zokors was amplified using *trn*L and ITS 1 primers, respectively. The sequence of these *trn*L primers were: *trn*L-*c*-F_*trn*L-*h*-R (*c*-F: CGAAATCGG TAGACGCTACG; *h*-R: CCATTGAGTCTCTGCACCTATC) [19]; ITS 1: ITS 5-F_ ITS 2-R: (5-F: GGAAGTAAAAGTCGTAACAAGG; 2-R: GCTGCGTTCTTCATCGATGC) [17].

## DNA extraction and PCR amplification

Total DNA was extracted from the stomach contents of 25 zokors according to the manufacturer's protocol of the E.Z.N.A.® soil DNA kit (Omega Bio-Tek, Norcross, GA, USA). Next, 1% agarose gel electrophoresis was used to detect the quality of DNA extraction. DNA samples were quality checked, and the concentration was determined using a NanoDrop 2000 spectrophotometer (Thermo Fisher Scientific, Wilmington, DE, USA).

Chloroplast *trn*L-*c*-F_ *trn*L-*h*-R and eukaryote ITS 1 primers were selected for PCR amplification. The raw materials used for PCR reaction were as follows: Transgen AP221–02:*Transstart* FastPFU DNA Polymerase, 20μL reaction system: 5 × *TransStart* FastPfu buffer (4μL), 2.5 mM dNTPs (2μL), forward primer (5μM) 0.8μL, reverse primer (5μM) 0.8μL, *TransStart* FastPfu polymerase 0.4μL,BSA 0.2μL, template DNA 10 ng, up ddH$_2$O to 20μL, 3 replicates for each sample. PCR reaction parameters for *trn*L-*c*-F;*trn*L-*h*-R were as follows: 3 min at 95˚C; 35× (30 s at 95˚C; 30 s at annealing temperature 56˚C; 45 s at 72˚C); 10 mins at 72˚C, 10˚C until halted. PCR reaction parameters for ITS 5-F_ ITS 2-R were as follows: 3 min at 95˚C; 35× (25 s at 94˚C; 30 s at annealing temperature 55˚C; 25 s at 56˚C); 10 mins at 72˚C, 10˚C until halted.

## Illumina sequencing of *trn*L and ITS 1 amplicons

Agarose gel electrophoresis was performed to verify the size of amplicons. Amplicons were then subjected to paired-end sequencing on the Illumina MiSeq PE300 sequencing platform at Majorbio Bio-Pharm Technology Co. Ltd. (Shanghai, China). The raw reads were deposited into the NCBI Sequence Read Archive (SRA) database (Accession Number: PRJNA753876).

## Sequencing data processing

Fastp (v0.19.6) [20] was used for quality control of the original sequences, and FLASH (v1.2.7) [21] was used for splicing. The reads were filtered with a tail mass value of 20 or fewer bases, and set a threshold is 50 bp. If the average mass value in the threshold was lower than 20, the rear bases were removed from the threshold. The reads were filtered with a tail mass value of 50 bp, and the reads containing N bases were removed. According to the overlap relationship between PE reads, pairs of reads were merged into a sequence, and the minimum overlap length was 10 bp. The maximum error ratio allowed in the overlap area of spliced sequences was 0.2, which let us remove the non-conforming sequences. The samples were differentiated according to the barcode and primers at both ends of the sequence, and the sequence direction was adjusted. The number of mismatches allowed by the barcode was zero, and the maximum number of primer mismatches was two.

The OTU clustering and ASV methods were used to compare the processed data and identify the differences in species annotation between this two methods. UPARSE7.0.1090 was used for OTU clustering, and bioinformation statistical analysis were generated for these OTUs at a 97% similar level. The NT_V20200604 database was selected, and the RDP Classifier2.11 was used for sequence classification annotation. Usearch7 was used for generating OTU statistics, Mothur1.30.2 was used for Alpha diversity analysis, Qiime1.9.1 was used to generate abundance tables for each taxonomic level, and the distance of beta diversity was calculated. Regarding the ASV method, based on the default parameters, the DADA2 [22] plug-in in the QIIME2 process [23] was used to de-noise the optimized sequence after quality control splicing. The representative sequences of ASV were classified and identified using the "NR/NT Collection" of GenBank database at NCBI https://blast.ncbi.nlm.nih.gov/Blast.cgi. Taxonomic annotation of ASV in the "Nucleotide Collection (NR/NT)" of the NCBI the GenBank database at NCBI was performed using the multi_blast method in QIIME2 (V2020.2).

The sequence consistency was 0.8, and the sequence coverage was 0.8. The subsequent data were analyzed on the online platform of Majorbio Cloud Platform.

Two indices, %RA (relative abundance) and %FOO (frequency of occurrence) [24, 25], were used to evaluate the food composition of zokors. %RA refers to the percentage of the occurrence frequency of the ASV of a certain food species out of the total occurrence frequency of all food ASVs. The calculation formula was as follows: $\%RO_i = (N_i/\sum N_i) \times 100\%$, $\sum N_i$ is the sum of the ASV occurrence times of all the foods of the animal. %FOO refers to the percentage of samples containing the ASV of a certain food out of the total number of samples, and the calculation formula was as follows: $\%FOO_i = (N_i/N) \times 100\%$, $N_i$ represents the number of samples with an ASV of class i food, and N is the number of effective samples [25].

## Results

### Flora of study area

The flora in the study area were surveyed during September 2020, October 2020, and April 2021. In the study area, after the project of returning farmland to forest and grass was implemented, the main tree species planted were *Picea crassifolia*, *Pinus sylvestris*, *Larix gmelini*, *Betula platyphylla*, *Armeniaca sibirica* and *Populus davidiana*. Common shrubs were *Hippophae rhamnoides* and *Sambucus adnata*. The common herbs were mainly weeds such as Asteraceae, Poaceae, Fabaceae, Rosaceae, and Apiaceae. See S1 Appendix for more details on specific plant species.

### Sequencing information statistics

HTS showed that significant food DNA was amplified from all gastric content samples. The traditional OTU clustering method was used resulting in 2,694,529 sequences. After denoising, 1,986,045 sequences were left, and 2,995 OTUs were obtained by clustering. Overall, 99.80% of OTUs were annotated as Eukaryota. In these Eukaryota's sequences, Fungi accounted for 67.01%, Viridiplantae was 31.32%, Metazoa was 1.62%, and unclassified was 0.05%. The ASV method was used, resulting in 3,514,769 sequences, and 1,852,644 valid sequences were obtained after DADA2 denoising, resulting in 4,657 ASVs. Overall, 99.81% of the effective ASV sequences were annotated to Eukaryota, among which Fungi accounted for 69.55%, Viridiplantae was 29.52%, Metazoa was 0.82%, and unclassified was only 0.11%. Studies have shown that zokors are purely herbivorous animals possessing a well-developed cecum for decomposing cellulose in plants [26],this study focused on Viridiplantae.

### OTU vs. ASV taxonomic analysis

The OTU clustering method and ASV method were both used to generate taxonomic annotations for the amplified feeding sequences, and the annotations of the two methods were compared at the order, family, genus, and species levels. We obtained 25 orders, 36 families, 94 genera and 150 species by OTU method. 25 orders, 36 families, 92 genera and 166 species were obtained by ASV method. At the order, family, and genus levels, there was a high degree of coincidence between the annotated information obtained by these two methods, but at the species level, the number of overlapping species from these two methods decreased, and the annotated information of these two methods was dramatically different. The ASV method showed improved diversity of food species compared with the OTU clustering method; that is, it could annotate more species. Therefore, the ASV method was selected for subsequent dietary analysis (Table 2).

**Table 2. Comparison of the annotation ability of the OTU clustering method with the ASV method.**

|  | OTU only | Shared | ASV only |
|---|---|---|---|
| Order | 2 | 23 | 2 |
| Family | 5 | 31 | 5 |
| Genus | 29 | 65 | 27 |
| Species | 77 | 73 | 93 |

## Pan/Core species analysis

The Pan species curves flat at the order and family levels but were still in an increasing stage at the genus and species levels. It is suggested that the sample size may did not fully reflect the food composition of zokor in this region, and a further study was needed to expand the sample size. Based on the Core species curve, with increasing sample size, at 4–5 samples, there were Core species at the order and family level, whereas the number of Core species at the genus and species level approached zero (S1 Fig).

## Venn diagram of zokor diet

In the seven study areas, there were four shared food families (Asteraceae, Poaceae, Rosaceae, and Convolvulaceae) (Fig 2a), and four shared genera (*Artemisia*, *Echinops*, *Cirsium*, and *Convolvulus*) (Fig 2b). The main plant genera were *Artemisia*, *Echinops*, and *Cirsium*. In terms of the food species of *E. cansus* and *E. smithii*, there are 17 families(Fig 2c) and 31 genera (Fig 2d) that they choose to feed on. The Venn diagram indicated that the dietary overlap of these two species was high, especially at the family level.

## Diet of zokors after the combination of two primers

After combining the results annotated by the HTS two primer pairs, the total food composition of 25 zokors was obtained. The diet of zokors was then evaluated by Relative Abundance (%RA) and Frequency of Occurrence(%FOO). Species with %RA less than 0.01% were excluded from our analysis for two reasons. First, because their %RA and their %FOO was too low, bias would be introduced if they were included in this analysis [27]. Second, the importance of rare food categories may be overestimated if an analysis is conducted by ordering the occurrence of a food species.

By combining %RA and %FOO, the zokor diet was evaluated at the family and genus levels. At the family level, the %RA of Asteraceae was 38.16%, and the %FOO was 100%, indicating that every zokor fed on Asteraceae (Fig 3a). The %RA of Poaceae was 22.52%, and the %FOO was 96%, indicating only one zokor did not feed on Poaceae. Therefore, Asteraceae and Poaceae play an extremely important role in the zokor's diet, followed by Rosaceae, Pinaceae, Brassicaceae, etc. The tendency of %RA and %FOO at the family level showed relatively good consistency (Fig 3a). At the genus level, the %RA of *Echinops* was 14.45%, which was the highest among all plant genera, but the %FOO was only 56%. The %RA and %FOO of *Littledalea* were 11.86% and 40%, respectively. The %RA of *Artemisia* was 9.85%, surprisingly, and the %FOO was the highest among all plant genera at 84%. In addition, some plant genera with relatively important values of %RA and %FOO included *Cirsium*, *Elymus*, *Potentilla*, and *Bupleurum* (Fig 3b). The detailed %RA and %FOO tables are shown in S1 Appendix. The detailed %RA food composition of each zokor (S2 Fig).

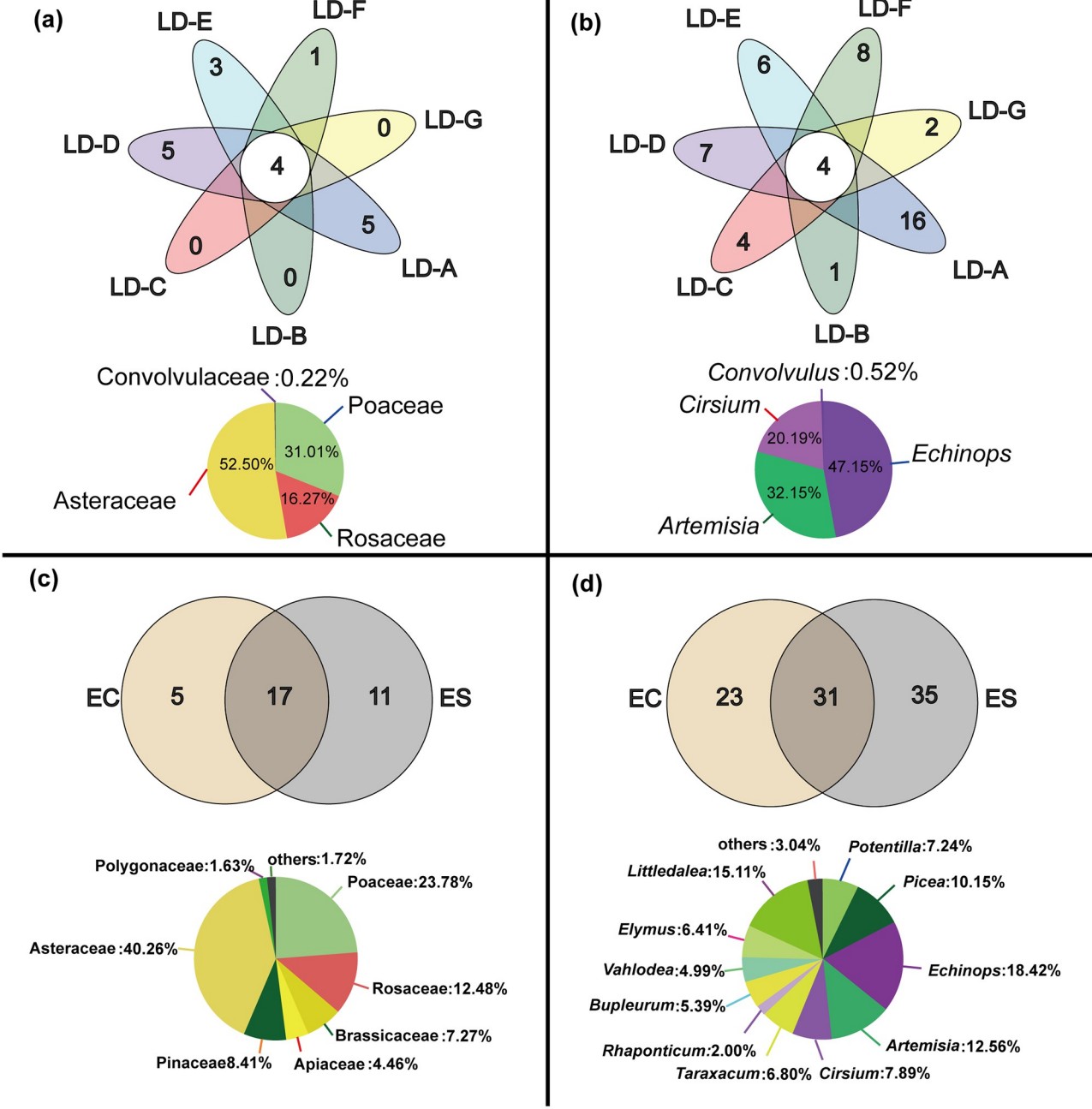

**Fig 2. Venn diagram of zokor diet. a** and **b** show the food species of the zokor at the family and genus level in the seven study areas respectively. The pie charts below show the food species of the zokor in all the study areas. **c** and **d** show the food species of the zokor at the family and genus level both species of zokors, respectively. The pie charts below shows the families and genera that both zokor species choose to feed on.

## Circos diagram of zokor's diet

A Circos diagram was next created to show the detailed correlation between the diets of zokors in different groups. Fig 4a displays the correspondence analysis of zokor diet among the seven plots at the genus level, and Fig 4b displays the correspondence analysis of the diet of two species of zokor at the genus level.

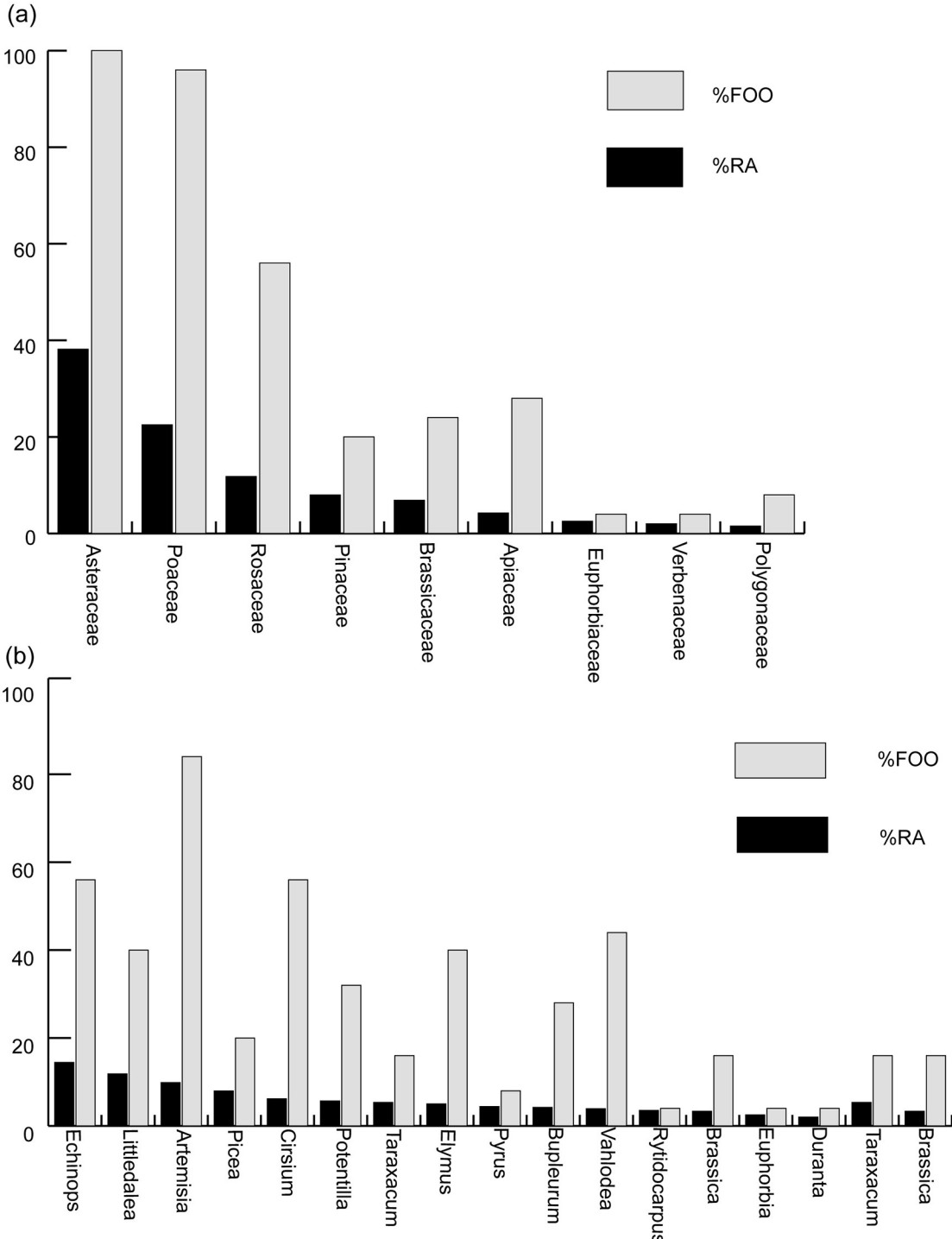

**Fig 3. %RA and %FOO of plant families and genera. a** and **b** respectively show that zokors mainly fed on families of the Poaceae, Asteraceae, and Rosaceae and genera of the *Echinops*, *Artemisia*, *Picea*, and *Cirsium*.

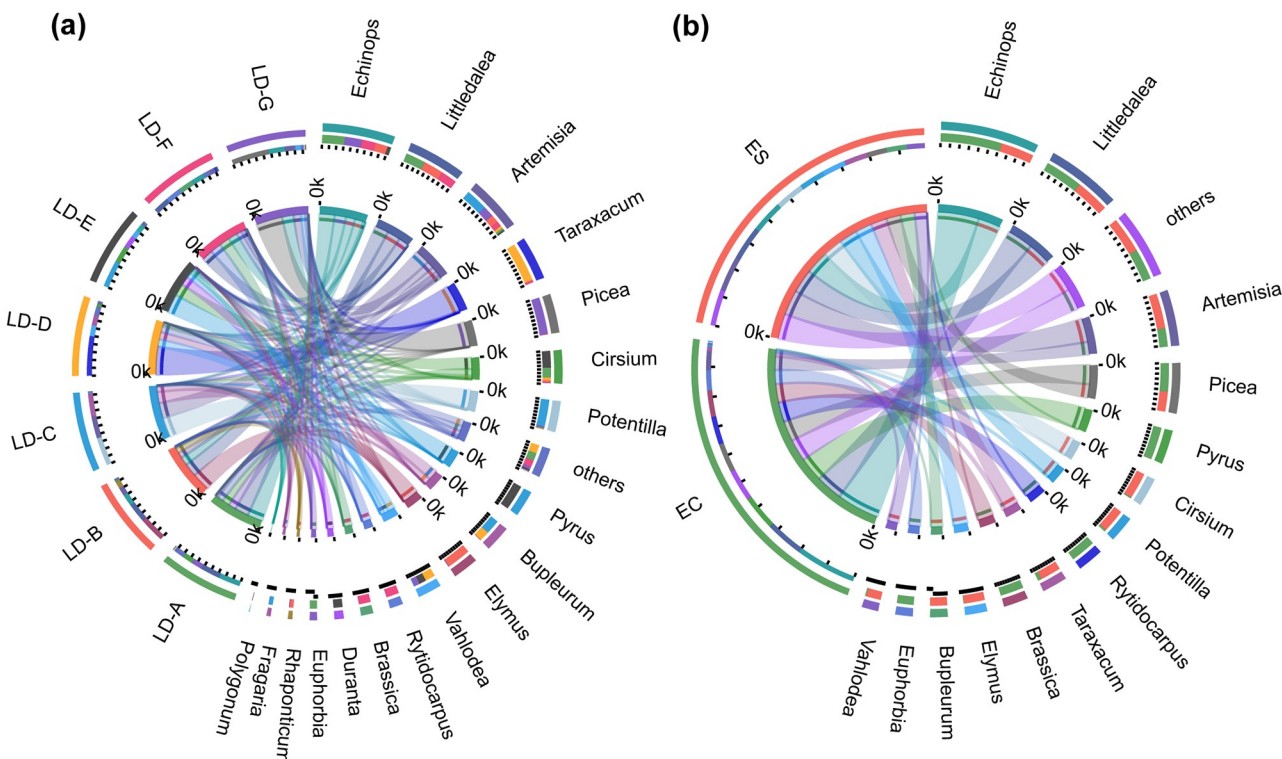

**Fig 4. Circos diagram of zokor diet at the genus level among zokors in different study areas and interspecies.** Genera with %RA < 5% are classified as others.

## Diversity analysis of zokor's diet

**Alpha diversity analysis.** As the Sobs index, Chao 1 index, and Ace index had almost no difference in the calculated diversity value, only the Sobs index was selected for analysis. The coverage index was always 1 because any ASV that only appeared once were removed from the sequence annotation, so the analysis of the coverage index was not carried out.

In seven study areas, the family level, the Sobs index of zokor diet ranged from 3.67 to 7.25 (mean 5.75), and the Shannon index ranged from 0.32 to 0.82 (mean 0.54), indicating low diet diversity. At the genus level, the Sobs index of zokor diet ranged from 9.33 to 14.40 (mean 11.35), and the Shannon index ranged from 0.36 to 1.10 (mean 0.86), also indicating low diet diversity. The Shannon evenness index reflects the evenness of the eating habits of zokor, which has the same trend as the Shannon index. There was no significant difference in the alpha diversity of zokor diet among the seven study areas.

The food Alpha diversity of *E. cansus* was slightly higher than that of *E. smithii* in terms of the Sobs index, Shannon index, and Shannon evenness index, but the sample size of *E. cansus* was smaller than that of *E. smithii* (9 < 16). The results suggest that *E. cansus* avoids interspecific competition with its relative *E. smithii* by adopting a more diverse diet. Due to its larger population, *E. smithii* consumes more available food resources in this area (Table 3).

**Difference test between groups of indices.** Kruskal-Wallis rank-sum test and false discovery rate (FDR) multiple test were next used to test the difference of diversity index between groups. It was found that there was no significant difference in alpha diversity of food composition of different study areas and interspecies (S3 Fig).

**Table 3. Alpha diversity index.**

| | Family | | | Genus | | |
|---|---|---|---|---|---|---|
| | Sobs | Shannon | Shannon evenness | Sobs | Shannon | Shannon evenness |
| LD-A | 6.80±0.98 | 0.67±0.31 | 0.34±0.16 | 14.40±4.45 | 0.92±0.43 | 0.34±0.14 |
| LD-B | 3.67±0.94 | 0.50±0.37 | 0.36±0.23 | 11.00±2.16 | 1.10±0.36 | 0.46±0.12 |
| LD-C | 5.00±1.63 | 0.32±0.39 | 0.17±0.19 | 9.33±2.62 | 0.36±0.38 | 0.15±0.14 |
| LD-D | 6.67±1.25 | 0.82±0.49 | 0.41±0.24 | 13.33±2.62 | 1.10±0.60 | 0.43±0.24 |
| LD-E | 6.33±1.25 | 0.73±0.16 | 0.40±0.06 | 10.00±1.41 | 0.80±0.12 | 0.35±0.05 |
| LD-F | 7.25±0.43 | 0.59±0.19 | 0.30±0.10 | 11.75±2.77 | 1.00±0.28 | 0.41±0.09 |
| LD-G | 4.50±1.66 | 0.43±0.27 | 0.31±0.21 | 9.75±3.70 | 0.73±0.45 | 0.30±0.17 |
| EC | 6.56±1.34 | 0.64±0.23 | 0.36±0.14 | 12.00±3.33 | 0.98±0.33 | 0.39±0.10 |
| ES | 5.44±1.80 | 0.55±0.41 | 0.31±0.22 | 11.31±3.79 | 0.80±0.51 | 0.32±0.20 |

**Correlation analysis of diet alpha diversity indices and body indices.** Next, the body indices of zokors were measured, including body weight and body length, and find correlations between diet diversity indices (Sobs index, Shannon index, Shannon evenness index) and other body indices. The results showed that the body weight and body length of zokors were slightly negatively correlated with the alpha diversity indices, and there was no significant difference (S2 Appendix). The correlation coefficient between the Sobs index and body weight was -0.382 ($P = 0.059$). The results seem to indicate that diet diversity may decrease with greater body size. Larger zokors appeared to be better at obtaining food than smaller ones, so the smaller ones must choose a variety of foods to meet its needs.

**Beta diversity analysis.**

- **Principal Component Analysis (PCA)** PCA results showed that almost all (22/25) of the samples were clustered together, with only EC-7, EC-4, and ES-13 deviating from most of the samples. These results may indicate that the plant species in the study area had low heterogeneity, and the zokor's diet in different study areas was mostly clustered together. Additionally, the results may indicate the interspecific competition between the two zokor species was not strong, and the food between two species was similar.

- **Principal Co-ordinates Analysis (PCoA)** Compared with PCA, PCOA could differentiate food samples from the zokor species ($P = 0.004$), but not from zokor species ($P > 0.05$) (Fig 5).

## Analysis of differences between groups

- **Analysis of zokor diet between study areas** Although 3 of the top 10 foods had significant differences in the study area, the difference is mainly due to the difference in the distribution of these food resources in different study areas. In *Picea*, for example, LD-G had a large *Picea crassifolia* nursery site, from which we had four zokors, while the rest of the study areas had little or no *Picea crassifolia*. These observations suggested that the zokor will first eat food near its habitat.

- **Analysis of interspecific diet** Among the top 10 food of the two zokor species, only *Crisium* was significantly different ($P = 0.036$). There were no significant differences in interspecific selection for most other foods, suggesting that they had the same preference for most plants. More *Crisium* was consumed by *E.smithii* than *E.cansus*. This plant is a kind of Asteraceae weed widely distributed in the study area, and the reason for the difference may need to be further analyzed from the food characteristics (Fig 6).

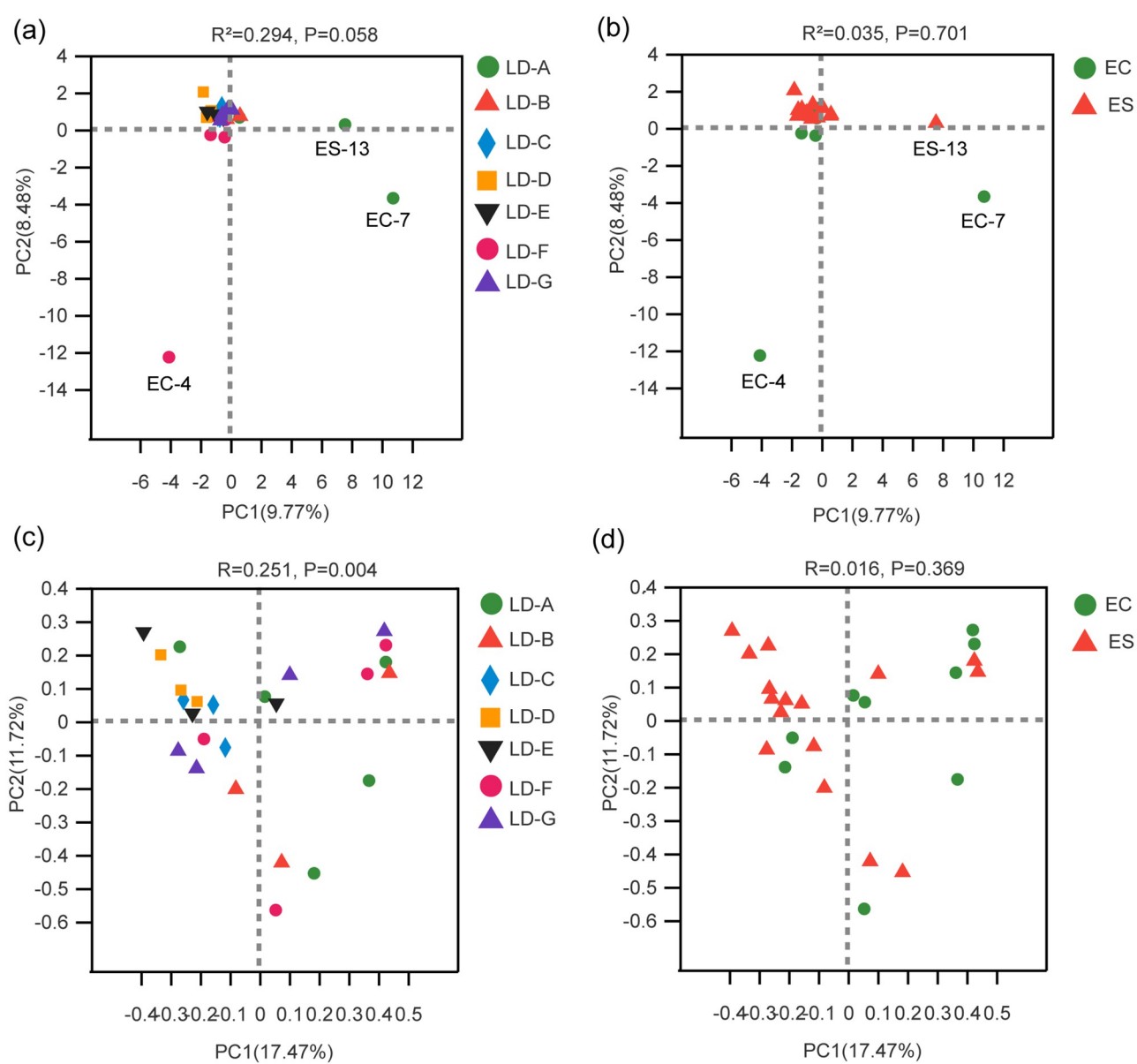

**Fig 5. PCA and PCoA plots of food composition of zokor at genus level. a** and **b** were PCA plots. **a** was the food PCA plot of the seven study areas, and **b** was the interspecific PCA plot of the two zokor species. **c** and **d** represent PCoA plots, **c** was the food PCoA plot of the seven study areas, and **d** was the interspecific PCoA plot of the two zokor species.

## Analysis of food selectivity

Through investigating the research area of the plant resources, a list of plants in the study area was compiled (S3 Appendix). The selectivity of zokors to plants was compared between plant species with %RA greater than 0.1%. Comparative DNA metabarcoding was performed on the plants of the genus for the zokor feeding ratio, and the selectivity index was calculated. Ivlev's selectivity index [28] was used to calculate the plant selectivity of zokor. $E_i = (r_i - p_i)/(r_i + p_i)$, where i is the number of plant species in the study area, $r_i$ refers to the proportion of i plants in the feeding composition, and $p_i$ refers to the proportion of i plant biomass to the total biomass

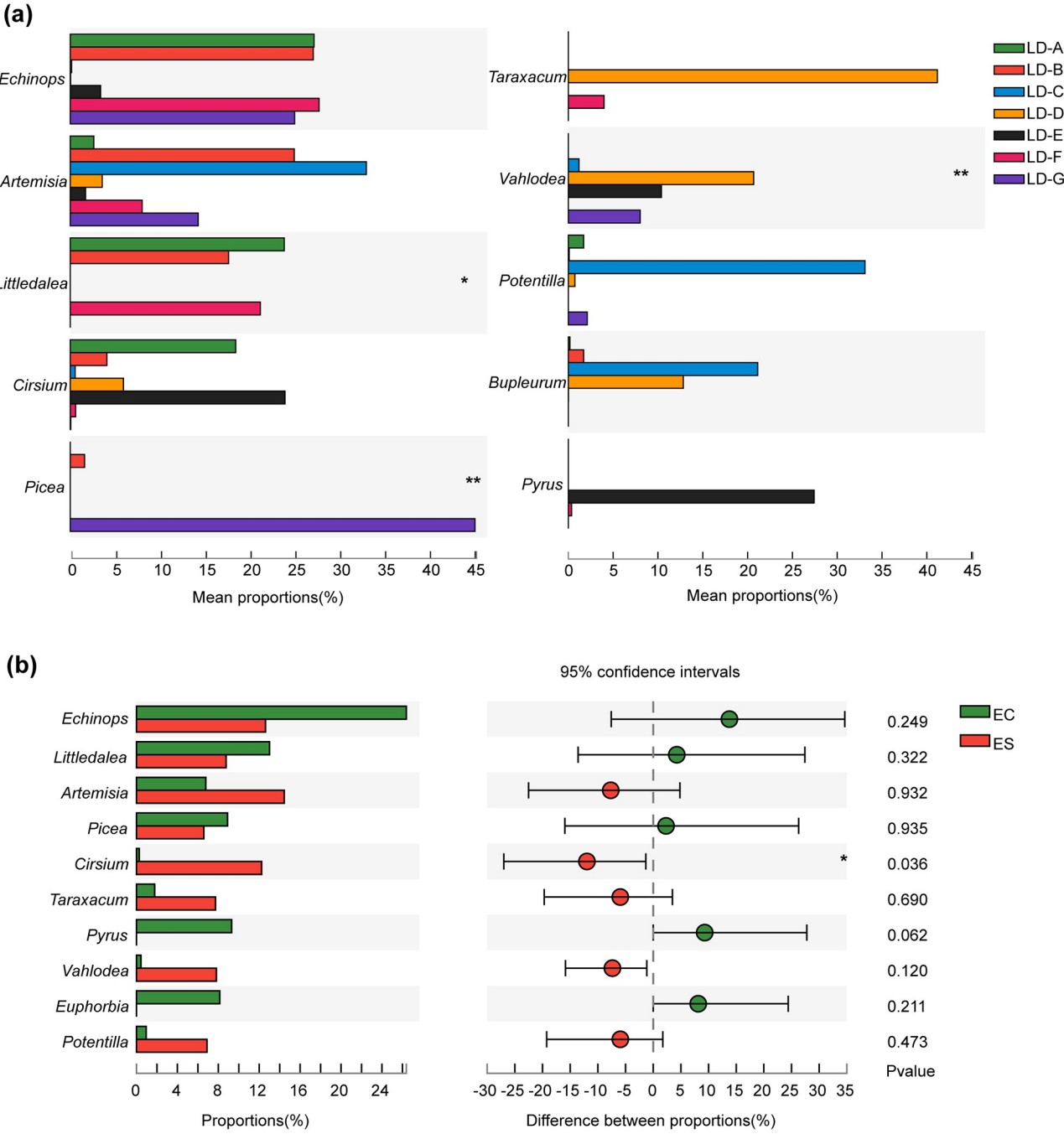

**Fig 6. Analysis of the difference between zokor species and different study areas. a** shows the top ten food species of the food composition %RA of zokor in seven study areas. **b** shows the top ten food species in %RA of interspecific food composition of two zokor species.

in the plant survey quadrate. $E_i$ is between -1 and +1, where $E_i > 0$ indicates that the animal has a positive choice for the plant, $E_i < 0$ means negative selection, and $E_i = -1$ means no selection.

It can be seen that the zokor feeds positively on the genera *Echinops*, *Bupleurum*, *Crisium*, *Brassica*, *Calamagrostis*, *Medicago*, *Sanguisorba*, and other Poaceae, which accounts for

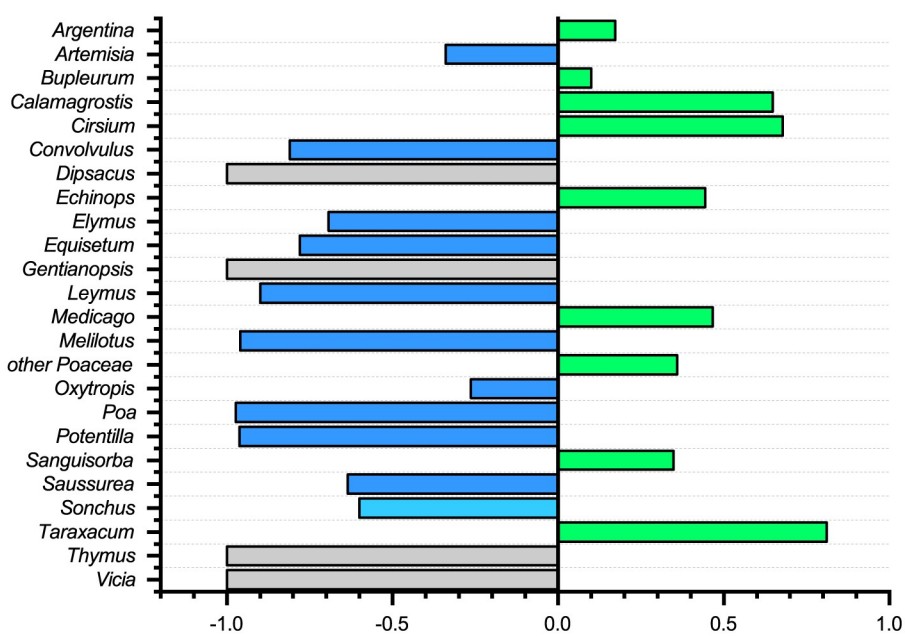

**Fig 7. Food selection diagram of zokor. Green** represents positive selection; **Blue** represents negative selection; **Grey** means no selection.

21.87% of the plant resources in this area, but 53.22% of the zokor's diet. As for the genera *Elymus*, *Leymus*, *Artemisia*, *Poa*, *Potentilla*, and *Convolvulus*, they were widely distributed in this area, accounting for 74.15% of the plant resources but only represented 16.43% of the zokor's diet (Fig 7).

## Food type analysis

Through a comparison of the annotated food list of zokors and the stored roots dug out from the caves of zokor, the plants fed upon by zokors could be classified in terms of their life forms and feeding parts.

**Plants classified in terms of their life forms.** Based on the life form of the plant and the abundance of that type of plant in the study area, they could be divided into annual herbs (AH), perennial herbs (PH), arbors (A), and other types. Other types include shrubs (S), biennial herbs (BH), floating algae (FA), and ferns (F) (Fig 8).

**Plants classified in terms of their feeding parts.** Plants in the zokor's diet could also be classified according to the plant parts that zokors feed on. Zokors mainly feed on taproots (54.60%) and rhizomes (26.41%)(Table 4), which are rich in water and energy and thus can meet the high energy consumption requirements caused by the zokor's digging habits [26].

## Discussion

In this study, the diet of zokors was determined by DNA metabarcoding based on HTS, marking the first ever report on the food composition of *E. smithii*. The results suggest that DNA metabarcoding is feasible for studying the feeding habits of rodents that live entirely underground and feed mainly on plant roots. In the study area, *E. smithii* is sympatric with *E. cansus*, and both belong to *Eosplax*. Studies have shown that the diets of seven species of *Ctenomys* in Brazil have a high overlap [4]. Similarly, the results of the present study showed there was no

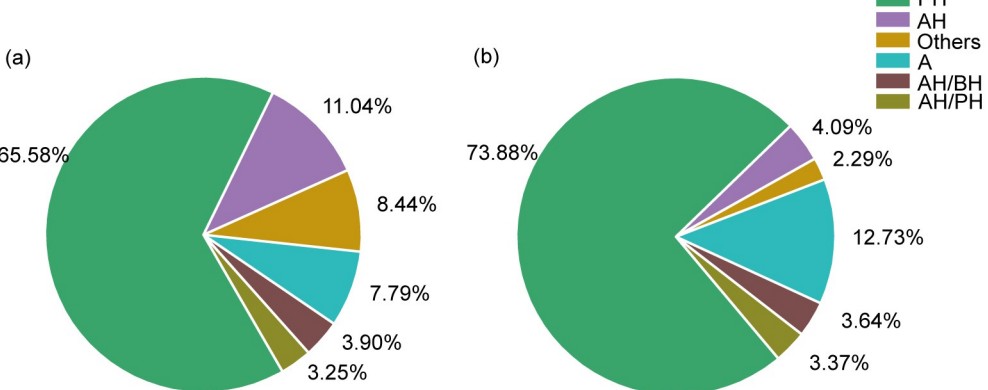

(a)    (b)

**Fig 8. Pie chart of plant feeding parts of the zokors. a** and **b** are the proportion of the number of plant species and the number of ASVs of the feeding type of zokor.

significant difference in food composition between *E. smithii* and *E. cansus*, with a Schoener niche overlap index of 0.69. In this area, two species of zokors mainly fed on the roots of perennial herbs, which mainly belonged to the genera *Echinops*, *Artemisia*, *Cirsium*, *Bupleurum*, and *Elymus*, as well as the roots of *Picea* and *Pyrus*. Since the samples in this study were the gastric contents of zokor and ITS primers were used to amplify the DNA of the gastric contents, many fungi were also amplified because the digestion degree of food in the stomach of mammals was the lowest, but the main groups were common plant pathological fungi, endophytic fungi, intestinal fungi. This fungus is either eaten by a zokor as it feeds on roots, or it is already present in the stomach, which may not be the result of the zokor's choice of food. This was also the case in the metabarcode feeding studies of lemmings' stomach contents [34].

Interestingly, ITS also amplified *Canis*. Since there were no stray dogs distributed in the study area, it was preliminarily speculated that it might be *Canis lupus*, which may mean that zokors ingested wolf feces in the process of digging and eating. Birds of prey (Falconiformes, Strigiformes) and other small predators (Canidae, Mustelidae, Felidae) are widely believed to be natural predators of the zokor. This view is however one-sided, since zokors rarely appear on the ground. The zokor's temporary herbivorous burrow is 10–30 cm below the ground, and its permanent cave is 50–210 cm below the ground [18]. Raptors, which hover high in the air and use their keen vision to spot prey, are unlikely to be able to spot zokors. The plateau zokor (*Eospalax baileyi*), which inhabits the Qinghai-Tibetan Plateau, does not appear in the local raptor diet, but pikas [35]. Small, slender carnivores, such as the *Mustela sibirica*, that would

**Table 4. Analysis of the feeding parts of plants.**

| Foraging type | Root type | Representative | Features | %RA |
|---|---|---|---|---|
| Tree root | Coniferous tree | *Picea asperata* | Fibrous and fatty [29] | 7.99% |
| | Other tree | *Pyrus, Diospyros* | Rich in fiber | 6.77% |
| Herbaceous root | Rhizome | *Elymus, Artemisia* | Reproductive part [29–31] | 26.41% |
| | Taproot | *Bupleurum, Taraxacum,* | Thick, watery, and nutritious [32] | 54.60% |
| | Tuber&bulbus | *Stachys geobombycis* | Watery and sugary [33] | 0.23% |
| | Fibrous root | *Setaria viridis* | / | 3.93% |
| Other | Rootless | FA,F | / | 0.06% |

be able to enter zokor burrows to prey on zokors, are more plausible predators, but research on this is lacking. Further research is needed to determine whether the presence of wolf feces in the stomach contents of zokors indicates that the wolf is a natural predator of the zokor.

Among the species of plants that were annotated in this study, Hydrodictyaceae (EC-2, ES-11, EC-6) and Polypodiaceae (ES-2) were unique in that they will only live in watery or humid areas. Both EC-2 and EC-6 were present in the LD-F plot, and both ES-11 and ES-2 were present in the LD-D plot, and their distances from streams were 180 m and 600 m, respectively. This suggests that the zokor is adapted to moist conditions, and that they travel considerable distances in search of food.

By analyzing the zokor's diet and comparing the plant types in the study area, we was found that the zokor is a typical opportunist, that is, it fed on most of the plants in the study area, which was consistent with many studies on the feeding habits of herbivores [5, 12, 15, 17]. Since most of the *Picea crassifolia*, *Pinus mongolicus*, and *Pinus tabulaeformis* in the study area (LD-A to LD-F) had wire nets around their roots to effectively prevent the biting of zokors, the zokors instead fed on the rhizomes of perennial herbs such as Poaceae and Asteraceae, which were widely distributed in the area. The zokors' food stores found in our plots consisted almost entirely of the asexual parts of plants.

It was autumn when zokors were collected for this study, which is the season for storing food. Two storehouses were found, one with five small pantries and one with three small pantries, which were neatly stuffed with plant roots. The contents could be mainly divided into gramineous weeds and forbs. Gramineous weeds accounted for 28.18%-34.23% of the raw weight, and forbs accounted for 64.84%-69.74% of the raw weight. Different from the storehouses of *Eospalax baileyi* [12], the zokor storehouses contained only the hypogeal parts of plants, not the aerial parts. It has been found that the zokor spends some of its time on the ground, which may allow it to feed on the aboveground parts of plants [36]. In autumn, the density of *Ctenomys* is positively correlated with the availability of plants with reserve organs [37]. Similarly, the zokor becomes significantly more active in autumn, actively searching for plant reserve organs (e.g.,rhizomes, tubers, taproots). These types of roots provide water, carbohydrates, fat, and protein. For example, dandelion roots contain 22.59% more carbohydrates than the aerial parts of the dandelion [32]. The root tuber of *Stachys geobombycis* contains around 23 g of sugar, 4.1 g of protein, 0.3 g of fat, and 72 g of water per 100 g [33] (Table 4).

Studies have shown that grassland management, such as grazing and mowing to reduce litter accumulation, can alleviate the negative impact of nitrogen deposition on plant diversity by reducing the asexual reproduction of dominant species [31]. The study area was dominated by perennial herbs, and the zokor's feeding on the asexual reproductive organs of these herbaceous plants may reduce the asexual reproduction of dominant species and indirectly slow down the decrease in plant diversity. In the study area, and in a larger area of woodlands, the zokor is considered a pest mainly because it gnaws on the roots of afforestation plants. The contribution of woodlands to soil and water conservation is well known, whereas the role of the zokor in the ecosystem is not well known. Only when the density of zokors is too high are they forced by intraspecific and interspecific competition to cause harm to forests. Most of the tree species in the study area were covered with wire nets, and the results and field survey showed that zokors did little harm to these trees. According to the literature, zokors can optimize the composition of meadow communities with appropriate density [13]. At the same time, as a basic species in the food web, zokors play a positive role in maintaining the stability of the ecosystem and promoting energy flow. Therefore, zokors should not be destroyed blindly and unilaterally. Although zokors did not cause great harm to the tree species in the study area, many pines outside the study area were destroyed by their feeding habits. For subsequent afforestation, we recommend the idea of protecting the selected species and tolerating

**Table 5. Comparison of zokor diet findings.**

| Researchers | Species | Method | Specimen number | Resources number | Feeding number |
|---|---|---|---|---|---|
| Wang QY et al. [13] | *E. baileyi* | Microhistology | 92 | 67 | 28 |
| Su JH et al. [14] | *E. baileyi* | Microhistology | 40 | 18 | 11 |
| Xie JX et al. [12] | *E. baileyi* | Dig the zokor's pantry | 57 | 66 | 59 |
| Wang J [11] | *E. cansus* | Captive feeding | 16 | 74 | 71 |
| Bazhenov YA [38] | *M. aspalax* | Dig the zokor's pantry | 3 | / | 11 |
| This study | *E. cansus & E. smithii* | DNA metabarcoding | 25 | 65 | 154 |

the existence of zokors to a certain extent, which is a win-win situation for the development of forestry economy and ecological restoration.

The findings of this study on the diet of zokor were compared with the findings of other researchers (Table 5). In this study, 65 species of plants belonging to 56 genera and 24 families were investigated in the study area, and 154 species of plants belonging to 80 genera and 32 families were detected by DNA metabarcoding technology (S4 Appendix). It could be directly observed that DNA metabarcoding greatly improved the identification of food species and improved our understanding of the zokor's diet. There were two possible reasons why the number of food species of zokors obtained by using DNA metabarcoding was higher than the number of plant species in the study area. Firstly, as typical burrowing animals, zokors mainly feed on the roots of plants, possibly taking in some soil in the process. In the study on the feeding habits of *Ctenomys*, it was found that 23 plant families and 58 molecular operational taxonomic units (MOTUs) could be recovered by extracting DNA from soil alone, and the diversity of plant families and MOTUS in soil samples was higher than that in the feces of *Ctenomys* [5]. Therefore, there may be false positives in the experimental results of the present study, that is, the actual variety of food resources consumed may be smaller than that recovered by DNA metabarcoding. Secondly, the plant species in the study area were investigated during autumn, when vegetation and trees were dying, so some species were not investigated.

Indeed, in a Pan species graph, more samples should be added for zokor diet analysis. Moreover, the study area was only 3 million square meters and had similar habitat types, whereas ideally samples should be collected over a larger geographical range and a longer time period to avoid bias [39]. Fortunately, samples were collected at different times over a sampling time span of nearly two months. In general, larger sample sizes and technical repetitions of a single sample (multiple extraction and sequencing) are certainly beneficial [40, 41]. However, due to the resulting increase in workload and cost, researchers cannot continuously increase the sample size and repetition. Instead, they need to find the best balance between sample size and biological and technical replication, taking into account the workload and cost [39]. The results of the present study are credible at the level of genus classification and can explain the dietary characteristics of zokors under natural conditions. Many other herbivorous animal feeding studies have been conducted at the level of family [42] and genus [17, 43]. In the study of feeding habits via DNA sequencing, there may be deviations/errors at every link, such as sample preservation, sample contamination, DNA extraction, PCR amplification, primer selection, library preparation, selection of sequencing platform, error removal, and sequence taxonomic assignment, which may affect the final result. Even so, the classification accuracy of diet studies based on HTS is much higher than that of traditional microscopic methods, and the workload is much less than that of microscopic methods.

## Supporting information

**S1 Fig. Pan/Core species analysis diagram.**
(TIF)

**S2 Fig. %RA food composition of each zokor.**
(TIF)

**S3 Fig. Alpha diversity of food composition of different study areas and interspecies (take the Sobs index).**
(TIF)

**S1 Appendix. Detailed %RA and %FOO tables.**
(XLSX)

**S2 Appendix. Correlation between food diversity index and body indexes of zokors.**
(XLSX)

**S3 Appendix. Flora of study area.**
(XLSX)

**S4 Appendix. Detailed DNA metabarcoding diet tables.**
(XLSX)

## Acknowledgments

We sincerely thank the Forestry Pest Control Station of Longde County, Ningxia Hui Autonomous Region, China, for its assistance in the collection of zokor samples. XXZ, YZ, XNN and CXH contributed to the study design. Sample collection and data analysis were completed by XXZ, YZ, XNN. Experiment was performed by XXZ, YZ and XZ. The first draft was written by XXZ. The revision suggestions for this paper were proposed by ZGX, XNN and CXH. All authors have read and approved the final manuscript.

## Author Contributions

**Conceptualization:** Xuxin Zhang, Xiaoning Nan, Chongxuan Han.

**Data curation:** Chongxuan Han.

**Funding acquisition:** Chongxuan Han.

**Investigation:** Yao Zou, Xuan Zou, Chongxuan Han.

**Methodology:** Xuxin Zhang, Yao Zou, Xuan Zou, Xiaoning Nan, Chongxuan Han.

**Project administration:** Chongxuan Han.

**Resources:** Xuxin Zhang, Xuan Zou, Xiaoning Nan.

**Software:** Xuxin Zhang, Yao Zou, Xuan Zou.

**Supervision:** Zhenggang Xu, Chongxuan Han.

**Validation:** Xuxin Zhang, Chongxuan Han.

**Visualization:** Xuxin Zhang, Zhenggang Xu.

**Writing – original draft:** Xuxin Zhang.

**Writing – review & editing:** Zhenggang Xu, Xiaoning Nan, Chongxuan Han.

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
