## [Decision Letter · Decision Letter 0]

4 Jan 2022

PONE-D-21-29707DNA metabarcoding Uncover the Diet of Subterranean Rodents in ChinaPLOS ONE

Dear Dr. Han,

Thank you for submitting your manuscript to PLOS ONE. After careful consideration, we feel that it has merit but does not fully meet PLOS ONE’s publication criteria as it currently stands. Therefore, we invite you to submit a revised version of the manuscript that addresses the points raised during the review process.

We look forward to receiving your revised manuscript.

Kind regards,

Feng ZHANG, Ph.D.

Academic Editor

PLOS ONE

https://journals.plos.org/plosone/s/file?id=ba62/PLOSOne_formatting_sample_title_authors_affiliations.pdf”

5. Please amend the manuscript submission data (via Edit Submission) to include author’s ,Yao Zou,Xiaoning Nan.

Reviewers' comments:

Reviewer's Responses to Questions

**Comments to the Author**

1. Is the manuscript technically sound, and do the data support the conclusions?

Reviewer #1: Yes

2. Has the statistical analysis been performed appropriately and rigorously? 

Reviewer #1: No

3. Have the authors made all data underlying the findings in their manuscript fully available?

Reviewer #1: Yes

4. Is the manuscript presented in an intelligible fashion and written in standard English?

Reviewer #1: No

5. Review Comments to the Author

Reviewer #1: Major comments:

1. The hypotheses of the present study were not proposed properly in the introduction. you give detailed introduction on methods, however, you are not aim to update the method used in this study.

2. A map to show the localities of samples is lacking in the present version.

3. It is unclear how did you identify these animals, and whether their live in different micro-habitat. It is better to provide pictures to show their differences in morphology and in habitat selection.

4. Authors mentioned ‘ The diversity of zokor’s diet differs greatly among different study areas’, but no figure was given to show these differences.

5. Authors mentioned ‘there was no significant difference between Family and Genus and among different species of zokor’ without give a picture or table. However, this should be one of the major conclusions of this study. I suggest to give pictures or well labeled tables to support your results in this section.

6. PCA results are quite important in this paper, it need to be placed in the major section of this ms.

7. Figures were not compiled properly, and present as separate small figures in this version. I suggest authors to compile small figures to one big plate by using adobe illustrator.

8. Legends for figures and tables are too simple to understand them correctly, abbreviations for figures and tables were not given in the current version.

9. I suggest you consult a person who is native in English to go through the ms.

6. PLOS authors have the option to publish the peer review history of their article (what does this mean?). If published, this will include your full peer review and any attached files.

Reviewer #1: No

---

## [Author Response · Author response to Decision Letter 0]

8 Feb 2022

Dear Reviewer：

We have revised the manuscript according to your suggestion. Thank you for your time in reviewing our manuscript.We have responded to each of your comments and hope that my response will answer your concerns.

---

## [Decision Letter · Decision Letter 1]

4 Apr 2022

DNA Metabarcoding Uncovers the Diet of Subterranean Rodents in China

PONE-D-21-29707R1

Dear Dr. Han,

We’re pleased to inform you that your manuscript has been judged scientifically suitable for publication and will be formally accepted for publication once it meets all outstanding technical requirements.

Kind regards,

Feng ZHANG, Ph.D.

Academic Editor

PLOS ONE

Additional Editor Comments (optional):

Reviewers' comments:

Reviewer's Responses to Questions

**Comments to the Author**

1. If the authors have adequately addressed your comments raised in a previous round of review and you feel that this manuscript is now acceptable for publication, you may indicate that here to bypass the “Comments to the Author” section, enter your conflict of interest statement in the “Confidential to Editor” section, and submit your "Accept" recommendation.

Reviewer #1: (No Response)

2. Is the manuscript technically sound, and do the data support the conclusions?

Reviewer #1: Yes

3. Has the statistical analysis been performed appropriately and rigorously? 

Reviewer #1: Yes

4. Have the authors made all data underlying the findings in their manuscript fully available?

Reviewer #1: Yes

5. Is the manuscript presented in an intelligible fashion and written in standard English?

Reviewer #1: Yes

6. Review Comments to the Author

Reviewer #1: Authors made great effort to improve this manuscript, I am happy with their revisions, so I recommend acceptance of this manuscript for publication.

7. PLOS authors have the option to publish the peer review history of their article (what does this mean?). If published, this will include your full peer review and any attached files.

Reviewer #1: **Yes: **Deyan Ge

---

## [Editor Report · Acceptance letter]

19 Apr 2022

PONE-D-21-29707R1 

DNA Metabarcoding Uncovers the Diet of Subterranean Rodents in China 

Dear Dr. Han:

I'm pleased to inform you that your manuscript has been deemed suitable for publication in PLOS ONE. Congratulations! Your manuscript is now with our production department. 

Kind regards, 

on behalf of

Dr. Feng ZHANG 

Academic Editor

PLOS ONE